# SPIDER: Multi-Layer Semantic Token Pruning and Adaptive Sub-Layer Skipping in Multimodal Large Language Models

## Abstract

Multimodal Large Language Models (MLLMs) face significant efficiency challenges that stem from two distinct yet coupled sources: data redundancy and computational redundancy. While most methods focus on data redundancy by pruning visual tokens from the output of the visual encoder or computing redundancy in LLM decoders using blockwise importance, the finer-grained inter-layer representation shifts and the distribution differences within the layers themselves have not been fully explored. In this work, we comprehensively investigate this dual-level inefficiency. We posit that intermediate layer tokens from vision encoders should be considered for effective visual token pruning, as semantic focus shifts across layers, with middle-layer tokens capturing more detailed object-centric information that deeper layers may abstract away. Furthermore, we reveal the differential contributions of Attention and FFNs across distinct LLM decoder layers. Building upon these discoveries, we propose **SPIDER**, a training-free framework that integrates multi-layer **S**emantic visual token **P**run**I**ng with an a**D**aptive sub-lay**ER** skipping mechanism. Experimental evaluations demonstrate that SPIDER consistently maintains strong performance across various MLLM architectures and reduction ratios. Notably, SPIDER achieves a reduction to $20\%$ in FLOPs for LLaVA-Next-7B, while preserving $98.7\%$ performance to the baseline.

## 1 Introduction

Multimodal large language models (MLLMs) integrate visual information with powerful large language models (LLMs), achieving wonderful performances on various complex tasks, such as image understanding (Bai et al., 2025), video understanding (Lin et al., 2024), and visual reasoning (Zhang et al., a). However, this integration introduces significant overhead, stemming from both data redundancy in the form of lengthy visual token sequences and computational redundancy within the large-scale LLM backbone. While visual token pruning has become a dominant strategy to tackle data redundancy (Chen et al., 2024b; Zhang et al., b; 2024a), the computational redundancy in processing the remaining tokens through the LLM decoder has received not enough attention. In this work, we argue that data redundancy and computational redundancy can be combined, and each requires a more fine-grained pruning strategy.

First, on the data redundancy front, most token pruners (Zhang et al., 2024a; b; Chen et al., 2024b) exclusively use features from the final layer. This overlooks a crucial semantic focus shift across encoder layers, impairing performance on fine-grained tasks like counting or localization. As visualized in Figure 1, while deep-layer features provide global context, they may not contain enough object-centric details that are rich in middle layers. Pruning solely based on final-layer features thus risks discarding parts of key objects.

Second, on the computational redundancy front, some methods focus on layer skipping. But most layer-skipping methods (Lawson & Aitchison, 2025; Csordás et al.; Raposo et al., 2024) are coarse-grained, indiscriminately skipping entire decoder blocks. Our analysis reveals that attention and Feed-forward Network (FFN) sub-layers contribute unequally to visual tokens. This finding motivates a more adaptive and fine-grained skipping strategy than treating blocks as monolithic units.

Based on these findings, we propose SPIDER, a novel, training-free framework to improve MLLM inference efficiency. It is built on two core mechanisms: **Multi-layer semantic token pruning (MSV-Prune)**: This strategy uses tokens from both deep and middle layers of the visual encoder for semantic clustering and similarity computation. **Adaptive sub-layer skipping (ASL-Skip)**: This strategy accumulates a skippability score to determine which retained visual tokens should perform layer skipping and at which layer to skip, and an offline sub-Layer contribution score (SLC) to decide which specific sub-layer (Attention or FFN) to skip in subsequent LLM decoder layers for skipped tokens. Our contributions are summarized as follows:

- We explore the semantic focus shift across vision encoder layers, and propose multi-layer semantic token pruning, considering middle and deep layer tokens.
- We quantify the fine-grained redundancy in attention and FFN sub-layers of MLLM decoders, and propose adaptive sub-layer skipping to decide whether a retained visual token should skip, when to skip, and ship which part of the layers.
- We propose SPIDER, a training-free framework combining token pruning and sub-layer skipping. It significantly reduces computational burden for various MLLM architectures while maintaining comparable performance.

## 2 RELATED WORKS

### 2.1 MULTIMODEL LARGE LANGUAGE MODELS (MLLMS)

MLLMs have rapidly advanced in vision–language understanding and generation (Comanici et al., 2025; Liu et al., 2024a). Notable systems such as LLaVA (Liu et al., 2024a), BLIP2 (Li et al., 2023a), and MiniGPT-4 (Zhu et al., 2023) can process prompts that integrate both text and images, enabling versatile multimodal interaction. Beyond general-purpose capabilities, MLLMs have been adapted to specialized domains, including affective computing (Li et al., 2024a) and medical reasoning (Zhang et al., 2025), highlighting their potential in high-stakes applications. Despite these advances, the computational demands of MLLMs remain substantial for both training and inference, especially in scenarios requiring fine-grained reasoning over high-resolution images (Zhang et al., 2024b;a). The challenge becomes more acute in video understanding (Maaz et al., 2023; Zhang et al., 2023; Lin et al., 2024), where temporal redundancy and extended frame sequences greatly inflate token counts. Such scalability and latency bottlenecks underscore the need for efficient MLLM architectures, motivating effective strategies for real-time, large-scale multimodal systems.

### 2.2 REDUCING REDUNDANCY IN MLLMS

While MLLMs have achieved remarkable success, their huge computational cost hinders scalable deployment. Existing inference optimizations primarily fall into two categories: token pruning and layer skipping.

Token pruning has gained wide attention due to the high redundancy of visual tokens, which dominate input sequences. Some efforts focused on compressing visual tokens into compact representations Chen et al. (2024a); Li et al. (2024b); Shang et al. (2024), but requiring additional training. Other training-free methods leverage text-visual attention within the LLM to identify less important visual tokens Chen et al. (2024b); Zhang et al. (b); Xing et al. (2024). However, studies like Vispruner (Zhang et al., 2024a), VisionZip (Yang et al., 2025), VTC-CLS Wang et al. (2024), and HiPrune Liu et al. (2025) argue against the sole reliance on text-visual attention due to positional bias, advocating for visual cues. A common limitation across most token pruning methods is their focus on pruning from the final output, neglecting the multi-layer semantic information inherent in the vision encoder.

Layer skipping addresses the inherent layer redundancy in MLLMs, as observed by ShortV Yuan et al. (2025). While some layer skipping approaches involve additional training Zeng et al. (2025); Suo et al. (2024); Elhoushi et al. (2024), our focus remains on training-free methods. ShortV (Yuan et al., 2025), for instance, identifies and replaces less effective layers with sparse versions where visual tokens remain frozen. However, these methods are not fine-grained enough since they often aggressively replace entire LLM layers for all visual tokens, overlooking token and sub-layer importance variance .

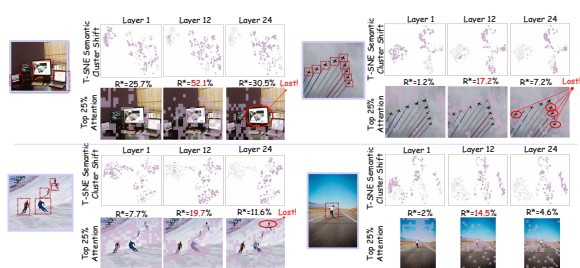 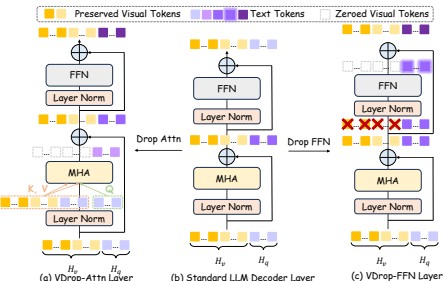

Figure 1: Visualizations to show the importance of mid-layer features for fine-grained token pruning. We highlight the top 25% most attentive tokens (in purple) from various layers of the CLIP vision encoder, together with their t-SNE embeddings. Mid-layer tokens exhibit a higher key object coverage ratio ($R*$), indicating a stronger focus on crucial object-centric details compared to deeper layers, which tend to capture broader global semantics.

Figure 2: Visualization of sub-layer skipping, where we visualize either (b) skipping attention sub-layer module or (c) FFN sub-layer module. The sub-layer skipping is achieved via replacing the standard LLM decoder block with corresponding sparse blocks, so that visual tokens are blocked in attention or FFN modules.

Different from previous works, we use the semantic information from middle and deep layer tokens for token pruning. Besides, we systematically analyze the contribution difference of attention and FFN modules in the LLM decoder and adopt a more fine-grained sub-layer skipping mechanism.

In this work, we propose SPIDER, a novel, training-free framework that synergistically tackles both token and layer redundancy with fine-grained adaptivity. Unlike previous token pruning methods, we explicitly consider the semantic differences between middle and deep layers of the vision encoder. We leverage a multi-layer semantic clustering approach by incorporating middle-layer tokens to achieve more comprehensive and effective token pruning. Furthermore, diverging from coarse-grained layer skipping strategies, SPIDER introduces a sub-layer skipping mechanism. This allows us to adaptively determine whether a retained visual token should skip computation, when to skip, and critically, which specific sub-layer (Attention or FFN) within the LLM decoder to skip, based on a nuanced understanding of their individual contributions.

## 3 KEY FINDINGS

### 3.1 SEMANTIC SHIFT ACROSS VISION ENCODER

We visualize the semantic focus shift and the distribution of the top 25% attentive tokens across CLIP vision encoder layers in Figure 1. t-SNE embeddings across layers show a gradual shift in semantic clustering, with middle layers bridging distinct visual concepts. Notably, middle-layer tokens focus more on key object regions, yielding higher key object coverage ratios $R*$ compared to early and deep layers. Deep layers, while capturing broader scene regions, risk missing critical object information. This suggests that effective token pruning should leverage both middle- and deep-layer features, not solely the deepest layer.

### 3.2 SUB-LAYER REDUNDANCY IN MLLMS

To quantify sub-layer redundancy for specific tokens, we introduce two sparse layers: **VSkip-Attn** and **VSkip-FFN**. In the VSkip-Attn layer, visual tokens do not function as queries, acting solely as keys and values. In the VSkip-FFN layer, the visual token input to the FFN is directly bypassed.

We propose the **Sub-Layer Contribution (SLC) score** to quantify a sub-layer's impact on the model's final prediction. The SLC is calculated as the Kullback-Leibler (KL) divergence between the output logits of the original model and a modified one where a specific sub-layer's operation is selectively bypassed. Specifically, to measure the contribution, we simulate skipping a sub-layer (e.g., attention or FFN) for a subset of tokens $X$ at a given layer $i$ by preventing their hidden states from being updated by that sub-layer. We then define $\mathbf{SLC}_{i,\text{attn}}^{X}$ and $\mathbf{SLC}_{i,\text{ffn}}^{X}$ as the KL divergence

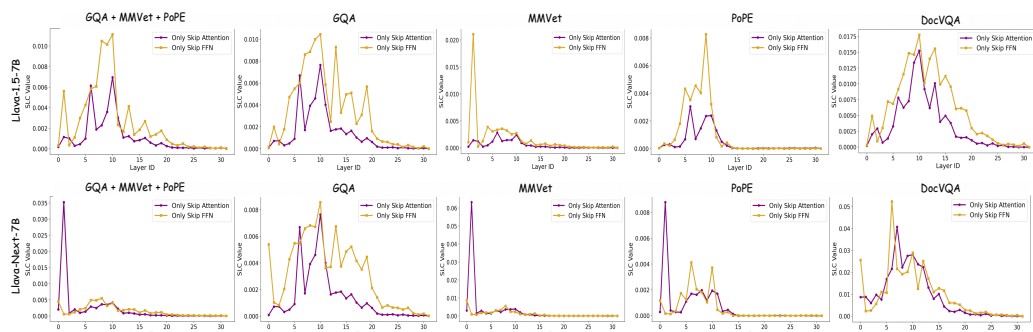

Figure 3: The sub-layer contribution scores (**SLC**) of LLaVA-1.5-7B and LLaVA-Next-7B. Lower **SLC** values indicate a weaker influence of the corresponding attention or FFN sub-layer on the specified tokens. Skipping transformations of visual tokens in such low-impact sub-layers yields minimal divergence from the original model's output distribution, and the **SLC** distribution is very different across various benchmarks.

when the self-attention or FFN sub-layers are skipped for tokens $X$ in layer $i$, respectively. A lower **SLC** value indicates less influence on the final output, making the sub-layer a stronger skip candidate. For each layer, we take $\mathbf{SLC}_i^X = \min\left(\mathbf{SLC}_{i,\mathrm{attn}}^X, \mathbf{SLC}_{i,\mathrm{ffn}}^X\right)$ to identify the least impactful module, replacing modules in ascending **SLC** order with corresponding sparse layers. Appendix A.2 provides a detailed sub-layer replacement order list.

To ensure robustness, we compute **SLC** offline across three diverse benchmarks. As shown in Figure 3, the resulting SLC distributions exhibit significant variance across different models and datasets, validating our approach of averaging them to obtain a statistically meaningful, generalized score. Another key finding is that $\mathbf{SLC}_{i,\mathrm{attn}}^X$ is generally lower than $\mathbf{SLC}_{i,\mathrm{ffn}}^X$. This suggests that for visual tokens, the attention sub-layer is often more redundant than the FFN, making it a preferable target for skipping.

## 4 METHOD

Our training-free framework is illustrated in Figure 4. The input image first goes through **multi-layer semantic visual token pruning** to reduce data redundancy, then the retained tokens are fed to the LLM decoder for **adaptive sub-layer skipping**. Details are as follows.

### 4.1 MULTI-LAYER SEMANTIC VISUAL TOKEN PRUNING

The retained visual tokens comprise two subsets: a fraction $r$ of anchor tokens $\mathbf{T}_v^{\mathrm{anc}}$ selected via attention sorting, and complementary tokens $\mathbf{T}_v^{\mathrm{cmp}}$ selected under the involvement of middle layer tokens into semantic clustering and sorting. Then $\mathbf{T}_v^{\mathrm{anc}}$ and $\mathbf{T}_v^{\mathrm{cmp}}$ are concatenated in spatial indices order and fused with text tokens $T_q$ for LLM decoding.

#### 4.1.1 ATTENTION-BASED ANCHOR TOKENS

To address the high redundancy in visual inputs, we first perform a saliency-based token pruning step. Drawing inspiration from findings that visual encoder attention is a reliable indicator of patch importance Zhang et al. (2024a), we use the visual encoder's self-attention scores to select a compact set of important tokens. We average the self-attention matrix $\mathbf{A}$ over all heads to obtain $\mathbf{a}_v \in \mathbb{R}^n$, where for CLIP-like encoders $\mathbf{a}_v$ is the [CLS] row, and for encoders without [CLS] it is the mean attention each patch token receives. A dynamic threshold $\tau$ selects anchor tokens to meet a budget of $n \times R \times r$ tokens, where $R \in (0, 1)$ is the overall retention ratio and $r \in (0, 1)$ is a hyperparameter defining the proportion of anchor tokens within the retained set.

$$\tau = \min\{t \mid |\{a_i^v \geq t\}| \leq n \times R \times r\}, \quad \mathbf{T}_v^{\mathrm{anc}} = \{t_i^v \in \mathbf{T}_v \mid a_i^v \geq \tau\}. \tag{1}$$

Figure 4: **Illustration of SPIDER**. We begin by token pruning using the semantic information from the middle and last layers. The retained tokens are fed to the LLM for adaptive sub-layer skipping.

### 4.1.2 MULTI-LAYER SEMANTIC CLUSTERING-BASED COMPLEMENTARY TOKENS

Relying solely on foreground-centric anchor tokens ($\mathbf{T}_v^{\text{anc}}$) risks losing crucial background context. To mitigate this, we propose a coarse-to-fine strategy to select a set of complementary tokens ($\mathbf{T}_v^{\text{cmp}}$). $\mathbf{T}_v^{\text{cmp}}$ are obtained by clustering non-anchor tokens into $K$ groups, computing their multi-layer similarity scores, selecting the $N_k$ lowest-scoring tokens from each cluster, and aggregating them.

**Coarse-grained Semantic Clustering.** A naive redundancy removal on non-anchor tokens is suboptimal, as populous but uniform background regions would exhaust the selection quota, displacing smaller yet unique semantic areas. To address this, we apply K-means clustering to the high-level features $\mathbf{F}_v^L$ of non-anchor tokens, partitioning them into $K$ semantic clusters $\{C_k\}_{k=1}^K$. The total budget for complementary tokens, $N_{\text{cmp}} = n \times R \cdot (1 - r)$, is then allocated proportionally to each cluster as a quota $N_k$, ensuring all semantic groups are represented.

$$N_k = \max\left(1, \text{round}\left(N_{\text{cmp}} \cdot \frac{|C_k|}{\sum_{j=1}^K |C_j|}\right)\right) \tag{2}$$

**Fine-grained Multi-view Pruning.** Within each semantically homogeneous cluster $C_k$, we select the most informative $N_k$ tokens by leveraging hierarchical features. We observe that middle-layer features ($\mathbf{F}^M$) capture fine-grained object-centric details, whereas last-layer features ($\mathbf{F}^L$) encode global semantics, as analyzed in Section 3.1. Middle layer is the $12^{th}$ layer and deep layer is the $24^{th}$ (final) layer. To select a complementary set, we compute a multi-layer similarity score $\mathbf{S}_{ij}$ for any pair of tokens $(i, j)$. This score considers their similarity at both middle and deep layer feature levels (multi-layer intra-cluster similarity), and their similarity to the already-selected anchor tokens:

$$\mathbf{S}_{ij} = \underbrace{\left(\text{sim}(\mathbf{F}_i^L, \mathbf{F}_j^L) + \text{sim}(\mathbf{F}_i^M, \mathbf{F}_j^M)\right)}_{\text{Multi-layer Intra-Cluster Similarity}} + \underbrace{\left(\max_{p \in \mathbf{T}_v^{\text{anc}}} \text{sim}(\mathbf{F}_i^L, \mathbf{F}_p^L) + \max_{p \in \mathbf{T}_v^{\text{anc}}} \text{sim}(\mathbf{F}_j^L, \mathbf{F}_p^L)\right)}_{\text{Similarity to Anchor Tokens}} \tag{3}$$

where $\text{sim}(\cdot, \cdot)$ is the cosine similarity. For each cluster, we compute similarity scores for all tokens, retain the $N_k$ lowest-scoring ones, and aggregate them across clusters to form $\mathbf{T}_v^{\text{cmp}}$.

## 4.2 ADAPTIVE SUB-LAYER SKIPPING

Even after token pruning, significant computational redundancy persists within the MLLM decoder. While prior work has explored coarse-grained layer skipping, this often incurs performance degradation. We observe that sub-layers (i.e., self-attention and FFN) within each decoder block contribute unequally to the final output (Fig. 3). This motivates our fine-grained, *adaptive sub-layer skipping* policy, which dynamically bypasses less critical sub-layers for specific tokens. The decision process is guided by a hybrid mechanism, combining an online, per-token skippability score with an offline, per-layer sub-layer contribution score.

### 4.2.1 ONLINE SKIPPABILITY SCORE

To determine *if* and *at which layer* a token should perform sub-layer skipping, we compute an online skippability score $\mathbf{S}_{sa}(i, \ell)$ for each visual token $i$ at each layer $\ell$. It consists of two aspects:

---

**Algorithm 1** Pseudocode for SPIDER

---

**Require:** $x, \mathcal{E}, \mathcal{D}, R, r, T_{\text{skip}}, w_1, w_2$
**Ensure:** $y$
1: // **Stage 1: Multi-layer Visual Token Pruning**
2: Extract mid/last features: $\mathbf{F}^M, \mathbf{F}^L \leftarrow \mathcal{E}(x)$
3: Select anchor tokens $\mathbf{T}^{\text{anc}}$ (top-$nRr$ by attention score)
4: Cluster non-anchor tokens on $\mathbf{F}^L$; select representatives minimizing multi-layer similarity to anchors and within-cluster redundancy
5: $\mathbf{T}_{\text{input}} \leftarrow [\mathbf{T}_q; \text{concat}(\mathbf{T}^{\text{anc}}, \mathbf{T}^{\text{cmp}})]$
6: // **Stage 2: Adaptive Sub-Layer Skipping (layers** $1$ **to** $L/2$**)**
7: Precompute normalized SLC scores $\text{SLC}^{\text{Norm}}_{\text{Attn/FFN}}(\ell)$
8: **for** $\ell = 1$ to $L/2$ **do**
9:   **for** each active visual token $i$ **do**
10:      Compute retention: $\mathbf{R}(i, \ell) = \text{norm}(H(p_i)) + \text{norm}(\cos(h_v(i, \ell), q_t(\ell)))$
11:      Update skip accumulator: $S_{\text{skip}}(i) \mathrel{+}= w_1 \cdot \text{ReLU}(1 - \mathbf{R}) + w_2 \cdot \text{SLC}^{\text{Norm}}(\ell)$
12:      **if** $S_{\text{skip}}(i) \geq T_{\text{skip}}$ **then**
13:         Skip the sub-layer (Attn/FFN) with highest $\text{SLC}^{\text{Norm}}$ for token $i$ thereafter
14:      **end if**
15:   **end for**
16:   Forward layer $\ell$ with dynamic skipping
17: **end for**
18: Run remaining layers ($L/2 + 1$ to $L$) fully
19:
20: **return** $\mathcal{D}(\mathbf{T}_{\text{input}})$

---

**1. Intrinsic Information Entropy ($\mathbf{E}_{ii}$):** This measures the semantic uncertainty of a token's hidden state $h_v(i, \ell)$. We compute it as the Shannon entropy of the probability distribution $p(i, \ell) = \text{Softmax}(h_v(i, \ell) \cdot W_{\text{unembed}}^{\top})$, which results from projecting the hidden state into the vocabulary space $V$ via the language model's unembedding matrix $W_{\text{unembed}} \in \mathbb{R}^{V \times d}$. Low entropy indicates semantic convergence, making the token a candidate for skipping. We normalize the entropy to $[0, 1]$ to get $\mathbf{E}_{ii}(i, \ell)$.

**2. Image-Text Correlation Factor ($\mathbf{F}_{itc}$):** This is a common factor (Zhang et al., b; Chen et al., 2024b) that assesses a visual token's relevance to the text query. A token weakly correlated with the text context is more skippable. We measure this as the cosine similarity between the visual token's hidden state $h_v(i, \ell)$ and the averaged text context vector $q_t(\ell)$. The cosine similarity $\text{sim}(i, \ell)$ is also normalized to $[0, 1]$ to be $\mathbf{F}_{itc}$. We then define a *retention score* $\mathbf{R}(i, \ell) = \mathbf{E}_{ii}(i, \ell) + \mathbf{F}_{itc}(i, \ell)$. The final skippability score is its inverse:

$$\mathbf{S}_{sa}(i, \ell) = \text{ReLU}\left(1 - \mathbf{R}(i, \ell)\right). \tag{4}$$

A higher $\mathbf{S}_{sa}$ indicates a stronger signal for skipping.

### 4.2.2   OFFLINE SUB-LAYER CONTRIBUTION SCORE

Once a token is deemed skippable, we must decide *which* sub-layer (attention or FFN) to bypass. To inform this, we pre-compute a Sub-Layer Contribution (SLC) score via offline profiling. For each layer $\ell$, $\text{SLC}_{attn}(\ell)$ and $\text{SLC}_{ffn}(\ell)$ are the average KL-divergence between the original model's output and the output when the respective sub-layer is skipped via sparse layer replacement, evaluated over multiple multimodel benchmarks. A low KL-divergence implies the module is less critical. We normalize these to obtain module-level skippability scores, where a higher score means more skippable:

$$\text{SLC}^{Norm}_{module}(\ell) = 1 - \frac{\text{SLC}_{module}(\ell) - \text{SLC}_{module,min}}{\text{SLC}_{module,max} - \text{SLC}_{module,min} + \epsilon}, \quad module \in \{\text{Attn, FFN}\}. \tag{5}$$

At each layer $\ell$, the sub-layer with the higher $\text{SLC}^{Norm}$ score is designated as the target for skipping.

Table 1: Comparison of training-free MLLM efficiency methods. FLOPs Ratio denotes the proportion of FLOPs retained relative to the vanilla model. Best results are in **bold**.

| Method | TFLOPs | Ratio | VQAv2 | GQA | MMStar | MME | MMB | PoPE | MMVet | TextVQA | DocVQA | Acc. (%) |
|---|---|---|---|---|---|---|---|---|---|---|---|---|
| *LLaVA-1.5-7B (Upper Bound, All 576 Visual Tokens)* | | | | | | | | | | | | |
| Vanilla | 8.5 | 100% | 76.5 | 61.9 | 33.7 | 1510.7 | 64.1 | 85.9 | 31.1 | 58.2 | 21.5 | 100% |
| *Approximately 55% TFLOPs* | | | | | | | | | | | | |
| FastV ($K = 2, R = 50\%$) | 4.9 | 58% | 73.5 | 60.2 | 32.4 | 1475.6 | 64.3 | 84.0 | 29.8 | 57.2 | 17.3 | 95.54% |
| VTW ($K = 16$) | 4.7 | 55% | 66.3 | 55.1 | 32.8 | 1497.0 | 64.8 | 82.8 | 19.2 | 55.3 | 16.2 | 88.93% |
| ShortV ($N = 19$) | 4.7 | 55% | 75.7 | **60.9** | 33.3 | **1503.1** | 64.8 | 86.2 | 27.9 | 55.1 | 17.9 | 96.08% |
| VisPruner ($V = 288$) | 4.7 | 55% | 76.3 | 60.8 | 33.3 | 1477.9 | 63.7 | 86.3 | 30.3 | 57.8 | 20.7 | 98.60% |
| SPIDER ($N = 384, R_s = 55\%$) | 4.8 | 56% | **76.6** | **60.9** | **34.9** | 1498.5 | **64.9** | **86.6** | **30.7** | **57.9** | **20.9** | **99.87%** |
| *Approximately 25-30% TFLOPs* | | | | | | | | | | | | |
| FastV ($K = 2, R = 75\%$) | 2.6 | 30% | 74.3 | 56.6 | 30.8 | 1394 | 62.3 | 79.2 | 30.3 | 56.2 | 16.2 | 92.33% |
| ShortV ($l = 31$) | 2.1 | 25% | 56.1 | 47.7 | 29.3 | 771.5 | 56.1 | 58.5 | 17.2 | 35.7 | 9.2 | 67.76% |
| VisPruner ($V = 128$) | 2.3 | 27% | 75.8 | 58.2 | 32.9 | **1461.4** | 62.7 | 84.6 | 28.6 | 57.0 | 18.0 | 95.27% |
| SPIDER ($N = 144, R_s = 35\%$) | 2.2 | 26% | **76.0** | **58.6** | **33.7** | 1458.2 | **63.7** | **84.8** | **30.3** | **57.3** | **18.5** | **96.73%** |
| *LLaVA-NeXT-7B (Upper Bound, All 2880 Visual Tokens)* | | | | | | | | | | | | |
| Vanilla | 42.7 | 100% | 80.0 | 62.9 | 37.1 | 1519.0 | 67.1 | 86.3 | 38.5 | 59.6 | 68.4 | 100% |
| *Approximately 50% TFLOPs* | | | | | | | | | | | | |
| FastV ($K = 2, R = 50\%$) | 22.0 | 52% | 79.5 | 63.0 | 36.5 | 1482.0 | 66.3 | 86.5 | 36.8 | 58.1 | 59.0 | 97.10% |
| VTW ($K = 16$) | 21.8 | 51% | 75.6 | 55.8 | 37.6 | 1518.2 | 67.1 | 84.9 | 18.5 | 57.3 | 58.2 | 89.71% |
| ShortV ($l = 19$) | 21.6 | 51% | 78.8 | **63.4** | **37.8** | **1525.1** | **67.2** | 86.9 | 31.7 | 58.3 | 59.8 | 96.67% |
| VisPruner ($V = 1600$) | 21.8 | 51% | 79.9 | 62.5 | 37.3 | 1493.1 | 66.7 | 88.0 | **37.3** | 59.4 | 63.8 | 98.81% |
| SPIDER ($N = 1920, R_s = 50\%$) | 21.9 | 51% | **80.2** | 62.6 | 37.7 | 1510.5 | 66.6 | 88.2 | 36.6 | **59.7** | 64.6 | **99.11%** |
| *Approximately 20-25% TFLOPs* | | | | | | | | | | | | |
| FastV ($K = 2, R = 89\%$) | 8.5 | 20% | 71.9 | 55.9 | 32.1 | 1282.9 | 53.4 | 71.7 | 25.9 | 55.7 | 43.7 | 81.89% |
| ShortV ($N = 29$) | 9.7 | 23% | 58.6 | 49.7 | 30.4 | 884.5 | 51.2 | 56.6 | 21.5 | 36.5 | 14.1 | 63.56% |
| VisPruner ($V = 640$) | 9.1 | 21% | **79.8** | 61.4 | 36.5 | 1490.8 | 65.2 | 85.9 | **36.7** | 59.3 | 50.6 | 95.49% |
| SPIDER ($N = 710, R_s = 30\%$) | 9.0 | 21% | **79.8** | **61.8** | **37.3** | **1492.2** | **65.7** | **87.8** | 35.9 | **59.5** | 51.5 | **96.09%** |

## 4.3 Score Fusion and Cumulation

The final skipping decision integrates both online and offline scores. At each layer $\ell$ (up to the network's midpoint, $L/2$), we compute a fused score:

$$\mathrm{S}_{fuse}(i, \ell) = w_1 \cdot \mathbf{S}_{sa}(i, \ell) + w_2 \cdot \mathrm{SLC}^{Norm}(\ell), \qquad (6)$$

where $\mathrm{SLC}^{Norm}(\ell)$ is the score of the skippable module at that layer. This score is accumulated for each token: $\mathrm{S}_{skip}(i, \ell) = \mathrm{S}_{skip}(i, \ell - 1) + \mathrm{S}_{fuse}(i, \ell)$. Once $\mathrm{S}_{skip}(i, \ell)$ exceeds a threshold $T_{skip}$, token $i$ enters a "skip mode". For all subsequent layers, it will bypass the pre-determined sub-layer (attention or FFN) identified by the offline SLC scores. Tokens not in skip mode proceed normally. To facilitate understanding of SPIDER's two-stage design, we provide a concise pseudocode in Algorithm 1.

## 5 Experiments

### 5.1 Experiment Settings

**Models.** We evaluate our method on LLaVA-1.5-7B (Liu et al., 2024a) and the high-resolution LLaVA-NeXT-7B (Liu et al., 2024b). LLaVA-1.5 generates 576 visual tokens from 336×336 images. LLaVA-NeXT's sub-image partitioning strategy handles flexible resolutions, yielding 2,880 tokens for our evaluation, a 5× increase.

**Datasets.** We evaluate on various multimodel benchmarks, including GQA (Hudson & Manning, 2019), VQAv2 (Goyal et al., 2017), MME (Zhang et al., 2021), TextVQA (Singh et al., 2019), POPE (Li et al., 2023b), MMB (Liu et al., 2024c) , MMVet (Yu et al., 2023), MMStar (Chen et al., 2024c) and DocVQA (Mathew et al., 2021).

**Baselines.** We benchmark SPIDER against training-free efficient MLLM methods. Token pruning methods include FastV (Chen et al., 2024b), which prunes a fixed ratio $R$ of visual tokens post-layer $K$ based on attention scores; VTW (Lin et al., 2025), which discards all visual tokens after layer $K$; and VisPruner (Zhang et al., 2024a), which retains $V$ visual tokens from the vision encoder. Layer skipping method includes ShortV Yuan et al. (2025), which replaces $N$ LLM layers with its ShotV layers. Our SPIDER retains $N$ visual tokens after the pruning stage and then allows $R_s$ of them to skip certain sub-layer modules. All the methods are compared at similar FLOPs reduction ratios.

To perform an ablation study on SPIDER's components, we introduce additional specialized baselines. We evaluate our token pruning module, MSV-Prune, against other token pruners like Sparse-

VLM (Zhang et al., b) and VisionZip (Yang et al., 2025). To assess the efficacy of our layer skipping module, ASL-Skip, we benchmark it against ShortV Yuan et al. (2025) and naive strategies that uniformly skip all attention or FFN modules.

| Method | GQA | TextVQA | POPE | Acc. (%) |
|---|---|---|---|---|
| *Upper Bound, All 2880 Tokens (100%)* | | | | |
| LLaVA-NeXT-7B | 62.9 | 59.6 | 86.3 | 100.0% |
| *Retain 320 Tokens (↓ 88.9%)* | | | | |
| FastV | 55.9 | 55.7 | 71.7 | 88.47% |
| SparseVLM | 56.5 | 52.4 | 73.5 | 87.64% |
| VisionZip | 58.1 | 57.6 | 75.0 | 91.97% |
| VisPruner | 58.4 | 57.6 | 80.4 | 94.22% |
| MSV-Prune (**Ours**) | **59.0** | **58.1** | **83.3** | **95.93%** |
| *Retain 160 Tokens (↓ 94.4%)* | | | | |
| FastV | 49.8 | 51.9 | 51.7 | 75.39% |
| SparseVLM | 50.2 | 45.1 | 54.6 | 72.92% |
| VisionZip | 54.3 | 54.7 | 59.4 | 82.31% |
| VisPruner | 54.7 | 56.0 | 72.9 | 88.46% |
| MSV-Prune (**Ours**) | **55.1** | **57.1** | **73.3** | **89.45%** |

Table 2: Comparisons of our MSV-Prune, with other SOTA training-free token pruning methods. Best results are in **bold**.

| Method | R | MMStar | TextVQA | MME | Acc. (%) |
|---|---|---|---|---|---|
| *Upper Bound, All 576 Tokens (100%)* | | | | | |
| LLaVA-1.5-7B | 100% | 33.7 | 58.2 | 1510.7 | 100% |
| ShortV | 81% | 33.8 | 57.3 | 1503.3 | 99.42% |
| Skip All Attn | 82% | **34.1** | 51.1 | 1300.6 | 91.69% |
| Skip All FFN | 38% | 28.9 | 40.9 | 875.9 | 71.34% |
| Skip Partial FFN | 81% | 32.6 | 57.4 | 1483.1 | 97.84% |
| ASL-Skip (**Ours**) | 80% | 33.7 | **58.1** | **1504.9** | **99.81%** |

Table 3: Comparisons of our ASL-Skip with other training-free layer skipping/pruning methods. No visual token is pruned. Most of these methods are evaluated at around 80 % of the original TFLOPs for fair comparisons. "Random Skip" means random skipping certain sub-layer modules across all LLM layers. Best results are in **bold**. "R" denotes TFLOPs ratio.

| Method | MMB | MMB$^{CN}$ | POPE | SQA$^{IMG}$ | VizWiz | Acc. (%) |
|---|---|---|---|---|---|---|
| *Vanilla, 100% Tokens* | | | | | | |
| Qwen-2.5-3B | 77.3 | 73.0 | 87.0 | 80.4 | 68.3 | 100 % |
| *Approximately 35% TFLOPs* | | | | | | |
| FastV | 74.4 | 70.6 | 85.0 | 79.3 | 66.9 | 97.4% |
| VisionZip | 74.9 | 69.8 | 85.4 | **80.1** | 67.1 | 97.7% |
| HiPrune | 75.8 | 71.3 | 86.0 | 80.0 | 67.5 | 98.6% |
| SPIDER (**Ours**) | **76.0** | **71.7** | **86.1** | 80.0 | **67.8** | **98.9%** |
| *Approximately 25 % TFLOPs* | | | | | | |
| FastV | 72.4 | 69.2 | 82.7 | 79.6 | 66.2 | 95.9% |
| VisionZip | 73.5 | 67.4 | 84.6 | 80.0 | 66.3 | 96.2% |
| HiPrune | 74.0 | 69.3 | 84.7 | **80.3** | 66.5 | 97.1% |
| SPIDER (**Ours**) | **74.8** | **69.6** | **85.2** | 79.8 | **67.0** | **97.5%** |

Table 4: Results on Qwen2.5-VL-3B-Instruct. Best results are in **bold**.

| Method | POPE | TextVQA | MMB | Acc. (%) |
|---|---|---|---|---|
| *Upper Bound, All 576 Tokens (100%)* | | | | |
| LLaVA-1.5-7B | 85.9 | 58.2 | 64.1 | 100% |
| *Approximately 30% TFLOPs* | | | | |
| + PruMerge+ (Train) | 84.0 | 57.1 | 64.9 | 93.3% |
| + SPIDER (Train-Free) | 84.8 | 57.4 | 63.9 | 99.0% |
| + SPIDER (Train) | **85.0** | **57.6** | **64.4** | **99.5%** |

Table 5: Comparisons of training-aware modes. The TFLOPs are kept at a similar level around 30% for fair comparison, where 1/4 of visual tokens are preserved for PruMerge+.

## 5.2 MAIN RESULTS

We apply SPIDER to the classic LLaVA-1.5 and LLaVA-Next models and comprehensively compare against prior approaches. As shown in Table 1, SPIDER consistently matches or surpasses baselines across multiple benchmarks at similar or lower FLOPs, outperforming other training-free methods and achieving the best trade-off between efficiency and performance. Notably, even under a relatively aggressive FLOPs reduction ratio to 20%, SPIDER applied on LLaVA-Next-7B retains 98.7% of the overall performance, whereas skip-layer-based baselines such as ShortV degrade substantially. On challenging benchmarks such as MMVet and MMStar, which demand strong spatial awareness and reasoning capabilities, SPIDER maintains competitive accuracy, underscoring its robustness under high compression.

More specific, we also ablate the separate performances of our token pruning method MSV-Prune in Table 2 and layer skipping method ASL-Skip in Table 3. Under various extreme reduction rates, MSV-Prune exceeds SOTA pruning methods. This superiority also proves the effectiveness of introducing middle-layer tokens to assist pruning. ASL-Skip preserves overall MLLM performance more effectively than existing layer-pruning approaches at the same computational cost, showing the necessity of finer-grained sub-layer skipping.

To demonstrate the effectiveness of SPIDER on other MLLM architecture, we additionally apply it to Qwen2.5-3B-Instruct. As shown in Table 4, SPIDER achieves the highest accuracy in comparison to other SOTAs across various benchmarks at different TFLOPs ratios.

Beyond its training-free mode, SPIDER adapts to a training-aware framework for performance enhancement. Table 5 shows results after fine-tuning on LLaVA-1.5-7B using LoRA for 1 epoch with 665K instruction data, following the same training setting as LLaVA-PruMerge Shang et al. (2024). Notably, even the training-free SPIDER outperforms PruMerge+ in accuracy, and post-training refinement further elevates its performance.

## 5.3 ABLATION STUDY AND ANALYSIS

| Method | GQA | TextVQA | POPE | Acc. (%) |
|---|---|---|---|---|
| *Upper Bound, All 2880 Tokens (100%)* | | | | |
| LLaVA-Next-7B | 62.9 | 59.6 | 86.3 | 100.0% |
| *Retain 1920 Tokens* | | | | |
| *(a) Semantic Cluster* | | | | |
| w/o Semantic Cluster | 61.3 | 59.3 | 87.2 | 99.33% |
| Cluster Num =2 | 61.9 | 59.4 | 87.9 | 99.98% |
| Cluster Num =8 | 61.6 | 59.3 | 87.7 | 99.68% |
| Cluster Num =4* | **62.6** | **59.7** | **88.2** | **100.64%** |
| *(b) Multi-Layer Tokens* | | | | |
| w/o Middle Layer Tokens | 62.2 | 58.5 | 86.6 | 99.13% |
| only Middle Layer Tokens | 61.9 | 58.7 | 87.1 | 99.28% |
| w/ Middle Layer Tokens* | **62.6** | **59.7** | **88.2** | **100.64%** |
| *(c) Components of $S_{ij}$* | | | | |
| w/o Similarity with $T_v^{anc}$ | 62.5 | 59.3 | 87.8 | 100.20% |
| w/o Inter-Layer Similarity | 61.9 | 59.0 | 87.5 | 99.60% |
| cos → MSE | 62.3 | 59.2 | 87.7 | 99.99% |
| cos → KL | 62.2 | 59.1 | 87.4 | 99.77% |
| All Equipped* | **62.6** | **59.7** | **88.2** | **100.64%** |

Table 6: Ablation of our token pruning method, MSV-Prune, on LLaVA-Next-7B. Best results are in **bold**. '*' denotes our default setting.

| Method | MMStar | TextVQA | MME | Acc. (%) |
|---|---|---|---|---|
| *Upper Bound, All 576 Tokens (100%)* | | | | |
| LLaVA-1.5-7B | 33.7 | 58.2 | 1510.7 | 100% |
| *(a) Components of $\mathbf{S}_{fuse}$* | | | | |
| w/o $\mathbf{S}_{sa}$ | 33.5 | 57.6 | 1497.8 | 99.17% |
| w/o **SLC** | 32.7 | 58.0 | 1483.6 | 98.30% |
| All Equipped* | **33.7** | **58.1** | **1504.9** | **99.81%** |
| *(b) Accumulation Mode of $S_{skip}$* | | | | |
| w/o Accumulation | 33.4 | 57.8 | 1499.1 | 99.22% |
| w/ Accumulation* | **33.7** | **58.1** | **1504.9** | **99.81%** |
| *(c) Components of $\mathbf{S}_{sa}$* | | | | |
| w/o $\mathbf{E}_{ii}$ | 33.6 | 57.9 | 1501.0 | 99.52% |
| w/o $\mathbf{F}_{itc}$ | 33.5 | 57.7 | 1498.5 | 99.25% |
| $\mathbf{S}_{sa}$* | **33.7** | **58.1** | **1504.9** | **99.81%** |

Table 7: Ablation of our sub-layer skipping method, ASL-Skip. No visual token is pruned. Best results are in **bold**.

**Ablation on MSV-Prune (Table 6).** Under a fixed token budget (1920 tokens, $r = 0.7$), removing any component of MSV-Prune (semantic clustering, middle-layer tokens, or similarity scores $S_{ij}$) reduces accuracy, highlighting the importance of modeling semantic shifts across layers. Notably, discarding middle-layer tokens hurts fine-grained tasks like OCR (e.g., TextVQA) more severely, and using only middle-layer tokens to compute intra-cluster similarity helps extracting object-centric cues. Replacing cosine similarity in $S_{ij}$ with MSE or KL divergence also degrades performance: MSE is sensitive to non-semantic magnitude differences Bengio et al. (2013), and KL divergence may not be the most appropriate choice here, as the layer features do not naturally form valid probability distributions. The full model ("All Equipped") achieves the best average accuracy.

| Anchor Ratio | GQA | TextVQA | POPE | Acc. (%) |
|---|---|---|---|---|
| LLaVA-1.5-7B | 61.9 | 58.2 | 85.9 | 100.0% |
| 0 | 59.3 | 54.7 | 83.7 | 95.74% |
| 0.3 | 60.2 | 57.0 | 86.2 | 98.51% |
| 0.5 | 60.6 | 57.5 | 86.4 | 99.09% |
| 0.7* | **60.9** | **57.9** | **86.6** | **99.56%** |
| 1.0 | 60.8 | 56.9 | 86.1 | 98.74% |

Table 8: Ablation of anchor token ratio in our token pruning method, MSV-Prune. Best results are in **bold**. '*' denotes our default setting.

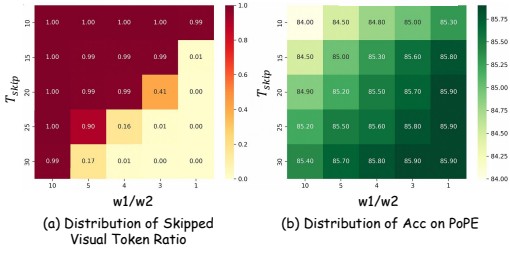

(a) Distribution of Skipped Visual Token Ratio  (b) Distribution of Acc on PoPE

Figure 5: Heatmap of token skipping proportion (left) and PoPE accuracy (right) for different $T_{skip}$ and $w_1/w_2$ settings on Llava-1.5-7B. No visual token is pruned.

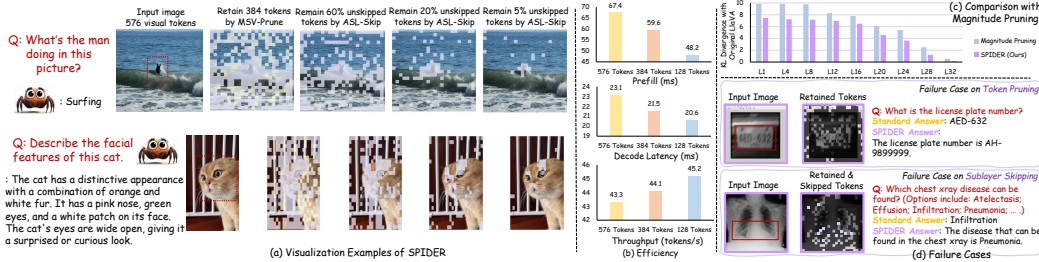

Figure 6: (a) Visualization of retained tokens by MSV-Prune and unskipped tokens by ASL-Skip; anchors in purple ($r = 0.7$), complementary tokens in yellow. (b) SPIDER efficiency on LLaVA-1.5-7B at varying token reduction ratios; key image regions relevant to queries marked by red dashed boxes. (c) KL divergence comparison with magnitude pruning across LLM layers. (d) Failure cases: red boxes highlight relevant areas; purple masks show skipped patch tokens during decoding.

**Ablation on Anchor Token Ratio in MSV-Prune (Fig. 8).** Varying the anchor ratio in MSV-Prune reveals that both insufficient and excessive anchoring degrade performance. Ratio=0.7 yields the best overall accuracy, indicating that balanced retention of anchor and complementary tokens maximizes efficiency without sacrificing quality.

**Ablation on ASL-Skip (Table 7).** When ablating $\mathbf{S}_{fuse}$ in (a), removing the $\mathbf{S}_{sa}$ component forces all visual tokens to follow the offline SLC list, while omitting **SLC** makes skipped tokens bypass entire subsequent layers, leading to higher performance variance and reduced robustness. In (b), the addition-based accumulation outperforms non-accumulative decisions, as tokens consistently deemed unimportant across consecutive layers are more reliably skip candidates than those judged by a single layer in isolation. In (c), both semantic uncertainty and image-text correlation are necessary for deciding if and when a token should start sub-layer skipping.

**Ablation on the Balance Between Token Skipping Ratio and Accuracy in ASL-Skip (Figure 5).** Heatmaps over $T_{\text{skip}}$ and $w_1/w_2$ show that aggressive skipping increases computational savings but can impair accuracy, while overly conservative skipping underutilizes efficiency gains. Appropriate parameter tuning ($w_1/w_2 = 3, T_{\text{skip}} = 20$) achieves a favorable trade-off.

## 5.4 VISUALIZATION & EFFICIENCY ANALYSIS

Figure 6(a) shows MSV-Prune retains semantically critical tokens, while ASL-Skip progressively skips less informative ones. Even at aggressive skip rates, core semantics are maintained, enabling accurate scene and object queries. Figure 6(b) shows fewer retained tokens significantly reduce prefill/decode latency and boost throughput, highlighting SPIDER's efficiency with minimal performance loss. We also compare our sub-layer skipping with magnitude pruning. For each layer $\ell$ and visual token $i$, we compute L1 norms of attention and FFN outputs, then average to get layer-wise scores $M_{\text{Attn}}(\ell)$ and $M_{\text{FFN}}(\ell)$. We prune the lower-magnitude sub-module, yielding a magnitude-pruned LLaVA. KL divergence between its per-layer logits and the original's measures post-pruning distortion. Figure 6(c) shows SPIDER achieves lower KL than magnitude pruning, confirming its superior layer-skipping with minimal degradation. Figure 6(d) shows SPIDER failure cases: (1) motion blur in license plates causes it to retain ambiguous regions and skip true digits; (2) in chest X-rays, critical tokens in the bilateral mid-to-lower lungs, especially perihilar, are allowed to skip sub-layers, causing ground-glass opacities to be missed and leading to misdiagnosis. These highlight challenges in token selection under severe image degradation and domain-specific reasoning.

## 6 CONCLUSION

We present SPIDER, a training-free framework that jointly performs multi-layer semantic visual token pruning and adaptive sub-layer skipping in MLLMs. By leveraging redundancy patterns across vision encoder layers and differentiating the roles of Attention and FFNs, SPIDER achieves substantial computational savings with minimal accuracy loss. Experiments across diverse MLLMs and benchmarks confirm the superiority of our method.

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

## A APPENDIX

### A.1 USE OF LLMS

We utilized large language models to assist with language editing and refinement to improve the clarity and readability of this paper.

### A.2 REPLACED SUB-LAYERS

For the selection of replaced layers, the **SLC** metric is computed based on a randomly sampled dataset comprising 150 cases, with 50 instances sampled equally from GQA, MMVet, and PoPE. Layers are then replaced, either with VSkip-Attn or VSkip-FFN modules, in ascending order of their **SLC** values. Specifically, replacement initiates from the lowest **SLC** value and proceeds towards higher values. Table 9 enumerates the layer IDs corresponding to the replaced components within the default SPIDER architecture.

Table 9: Replaced layers for different MLLM series and parameter scales.

| Model Series | Replaced Sub-Layers |
| --- | --- |
| LLaVA-1.5-7B | **Attention**: 25,27,28,30,23,26,22,24,21,0,20,3,18,4, 19,17,14,15,5,16,12,1,13,7,8,9,10; **FFN**: 31,29,2,11,6 |
| LLaVA-NeXT-7B | **Attention**: 28,29,27,30,23,24,25,22,21,26,20,19,18,17, 15,16,14,12,4,13,5,0,7,6,8; **FFN**: 31,1,2,3,11,9,10 |

## A.3 BENCHMARKS

VQAv2 (Goyal et al., 2017). VQAv2 evaluates visual recognition through open-ended questions on 265,016 MSCOCO images. Each image features at least three questions with adversarially balanced answers, preventing models from relying on statistical biases. We use the test-dev set (107,394 image-question pairs), where each question has 10 ground truth answers, scored by automatic evaluation metrics.

GQA (Hudson & Manning, 2019). GQA assesses structured understanding and reasoning in images. Beyond images and questions, it provides scene graph annotations (from Visual Genome) detailing objects, attributes, and relationships. Questions are generated via scene graphs, ensuring clear semantic paths. Evaluation uses accuracy on the test-dev set (12,578 image-question pairs).

TextVQA (Singh et al., 2019). TextVQA tests models' ability to recognize and integrate textual information (OCR) from images with natural language understanding. Images, primarily from Open Images v3, feature rich text (e.g., signs, packaging). Answers may require direct text extraction or contextual reasoning. Performance is evaluated on a validation set of 5,000 image-question pairs.

POPE (Li et al., 2023b). POPE measures hallucination in LVLMs by querying object presence in MSCOCO images. It assesses object hallucination degree. Evaluation uses the average F1 score across three sampling strategies on the test set (8,910 image-question pairs).

MME (Zhang et al., 2021). MME comprehensively evaluates multi-modal models' perceptual and cognitive capabilities across 14 subtasks. Perception includes OCR, coarse-grained (presence, count, position, color) and fine-grained (posters, celebrities, landmarks) recognition. All questions are binary. We report the perception score based on 2,374 image-question pairs.

MMB (Liu et al., 2024c). MMBench provides a comprehensive multi-modal evaluation with a three-level competence framework: two basic abilities (perception, reasoning), six specific capabilities, and twenty concrete tasks, all using multiple-choice questions. Both English (4,377 image-question pairs) and Chinese (MMBench-CN, 4,329 pairs) versions are utilized for evaluation.

MMVet (Yu et al., 2023). MM-Vet focuses on integrating diverse core vision-language capabilities. It defines six core capabilities (recognition, OCR, knowledge, language generation, spatial awareness, mathematics), combined into 16 specific tasks. ChatGPT assists evaluation, providing unified metrics for varied answer styles across 218 image-question pairs.

MMStar (Chen et al., 2024c). MMStar is a multi-modal benchmark designed to rigorously evaluate LVLMs by addressing issues of unnecessary visual content and data leakage in existing benchmarks. It comprises 1,500 human-curated samples that exhibit strong visual dependency, minimal data leakage, and require advanced multi-modal capabilities. MMStar assesses LVLMs across 6 core capabilities and 18 detailed axes, providing a purified and balanced evaluation to accurately measure true multi-modal gains and identify data leakage.

DocVQA (Mathew et al., 2021). DocVQA is a VQA dataset on document images, containing 50k questions over 12k+ documents. It emphasizes structural understanding, and often requires models to explore fine-grained textual and spatial cues to solve these information-dense settings.

## A.4 BOUNDARY SHIFT

To explore the extent of knowledge boundary drift caused by SPIDER, we conducted a fine-grained instance-level analysis on the POPE test set using LLaVA-1.5-7B with our SPIDER method (at 56% of original FLOPs). Accuracy improves from 85.9% to 86.6%, reflecting a net gain: the number of originally incorrect predictions corrected by SPIDER exceeds the number of originally correct ones flipped to wrong by 0.7%. Qualitative examples are provided in Fig. 7. This gain stems from SPIDER's retention of mid-layer tokens encoding object-centric regions, which enhances discriminative grounding and reduces false positives—particularly in hallucination-sensitive queries.

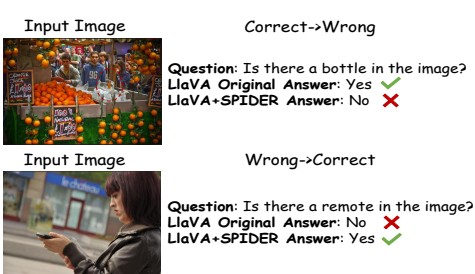

Figure 7: Instance-level visualization of boundary shift.

