# OpenReview forum: "SPIDER: Multi-Layer Semantic Token Pruning and Adaptive Sub-Layer Skipping in Multimodal Large Language Models"
_ICLR.cc/2026/Conference — Submitted to ICLR 2026_

### Official Review · Reviewer_v1Ke · 2025-10-26

**Soundness:** 3
**Presentation:** 3
**Contribution:** 3
**Rating:** 6
**Confidence:** 3

**Summary:**

This paper proposes SPIDER, a training-free framework for VLM token pruning. It introduces two complementary components: (1) Multi-layer Semantic Visual Token Pruning (MSV-Prune) : selects tokens using semantic features from both middle and deep layers of the vision encoder to preserve fine-grained object information. (2) Adaptive Sub-layer Skipping (ASL-Skip): dynamically skips attention or FFN sub-layers based on per-token entropy and image–text correlation, fused with precomputed sub-layer contribution scores. From result, the proposed methods performance well on most of benchmark with 80% reduction.

**Strengths:**

This paper provide a insight view of token pruning, as attention indeed shift between different layer, so maybe according to situation of each layer to choose token should be a promising idea.

This work provide comprehensive experiment on various benchmarks as well as hyperparameter anlyses.

**Weaknesses:**

1.The paper's writing needs improvement; the content too dense. The authors could perhaps provide a clearer description of the different components, such as following the data pipeline.

2.The method presented here is largely experimental, with a weak theoretical foundation.

3.If possible, author could experiment with more backbones to see how the proposed method generalizes.

**Questions:**

1. How stable are SLC scores across different datasets or tasks? Could a dynamic adjustment mechanism further improve performance? like classification task maybe much easier than qa.

2. Prior work has shown that token pruning methods may raise knowledge boundary drift. For example, instances that were previously answered correctly may become incorrect after pruning, and vice versa. Such inconsistencies could be problematic in applications where reliable responses to critical queries are essential.
Would it be possible for the authors to conduct an instance-level stability analysis to examine whether the proposed method causes any notable shifts in per-sample predictions or reasoning consistency?

---

> ### Author Response · Authors · 2025-11-18
> **Responses to Reviewer v1Ke**
>
> **Our manuscript has been updated, please check it.**
>
> **Q1.** The paper's writing needs improvement; the content too dense. The authors could perhaps provide a clearer description of the different components, such as following the data pipeline.
>
> **A1.** We sincerely thank the reviewer for this constructive suggestion. In our updated manuscript, we have added a clear algorithmic pseudocode (Algorithm 1) that outlines our MSV-Prune and ACL-Skip pipeline step-by-step, and improved the writing quality to make the method description clearer.
>
> **Q2.** The method presented here is largely experimental, with a weak theoretical foundation.
>
> **A2.** We appreciate the reviewer’s concern, but would like to clarify that our approach is not purely heuristic or empirical—it is grounded in statistical observation and reproducible analysis of token redundancy patterns in vision-language models. For example, the SLC is obtained based on averaging across various benchmarks of different tasks to make the score more stable and statistically meaningful.
>
>
> **Q3.** If possible, author could experiment with more backbones to see how the proposed method generalizes.
>
> **A3.** To demonstrate the effectiveness of SPIDER on other MLLM architecture, we additionally apply it to Qwen2.5-3B-Instruct and update the results in Table 4 (the text analysis is marked in magenta color). As shown in Table 4, SPIDER still achieves the highest accuracy in comparison to other SOTAs across various benchmarks at different TFLOPs ratios.
>
>
> **Q4.** How stable are SLC scores across different datasets or tasks? Could a dynamic adjustment mechanism further improve performance? like classification task maybe much easier than qa.
>
> **A4.** We thank the reviewer for this insightful question. To ensure cross-task robustness, we compute SLC scores offline on three diverse benchmarks: GQA (structured reasoning), POPE (factual grounding via object-presence queries), and MMVet (spatial reasoning, OCR, and math). As shown in Figure 3, SLC distributions vary across datasets. We average these scores into a single, generalized SLC profile, yielding a statistically stable prior that captures consistent token importance across tasks. This ensemble strategy requires no task-specific tuning and delivers consistent gains (Table 1), confirming its generalizability.
>
> Moreover, beyond the static SLC scores, we have explored a dynamic adjustment mechanism based on $S_{sa}$ that can dynamically adjust the skippability score of the retained visual tokens according to different tasks. Higher $S_{sa}$ indicates a stronger signal to start sub-layer skipping.
>
> **Q5.** Prior work has shown that token pruning methods may raise knowledge boundary drift. For example, instances that were previously answered correctly may become incorrect after pruning, and vice versa. Such inconsistencies could be problematic in applications where reliable responses to critical queries are essential. Would it be possible for the authors to conduct an instance-level stability analysis to examine whether the proposed method causes any notable shifts in per-sample predictions or reasoning consistency?
>
> **A5.** We appreciate the reviewer’s concern about prediction consistency after token pruning. We conducted a fine-grained instance-level analysis on the POPE test set using LLaVA-1.5-7B with our SPIDER method (at 56% of original FLOPs). Accuracy improves from 85.9% to 86.6%, reflecting a net gain: the number of originally incorrect predictions corrected by SPIDER exceeds the number of originally correct ones flipped to wrong by 0.7%. Qualitative examples are provided in Appendix A.4 Fig. 7 and highlighted in gray color. This gain stems from SPIDER’s retention of mid-layer tokens encoding object-centric regions, which enhances discriminative grounding and reduces false positives—particularly in hallucination-sensitive queries.

---

> > ### Comment · Reviewer_v1Ke · 2025-11-23
> >
> > Thank you for the comprehensive rebuttal. The clarifications provided effectively resolve my concerns. As my initial overall assessment was positive, I will maintain my score.

---

> > > ### Author Response · Authors · 2025-11-26
> > >
> > > Thank you for your thoughtful feedback and for considering our revisions. We greatly appreciate your time and constructive comments, which have helped improve the paper. We are glad that the clarifications were helpful.

---

### Official Review · Reviewer_H8DY · 2025-10-27

**Soundness:** 3
**Presentation:** 3
**Contribution:** 3
**Rating:** 6
**Confidence:** 3

**Summary:**

This paper addresses computational inefficiencies in Multimodal Large Language Models (MLLMs), focusing on data redundancy in vision encoders and computational redundancy in LLM decoders.

**Strengths:**

SPIDER does not require additional fine-tuning, making it practically attractive for deployment on pre-trained models.

**Weaknesses:**

1. While the proposed method demonstrates strong results on LLaVA-based models, the evaluation lacks coverage of a broader range of MLLMs, such as Qwen-VL. This raises questions about the generalizability of SPIDER across different model architectures.

2. The comparative experiments focus solely on training-free baselines. While this is aligned with SPIDER's training-free design, it remains unclear how the method would perform relative to training-based approaches.

3. Although the training-free nature of SPIDER is advantageous for deployment efficiency, it is worth investigating whether the method could be extended into a training-aware framework, potentially allowing learnable pruning or adaptive skipping to further enhance performance.

**Questions:**

1. Although the training-free nature of SPIDER is advantageous for deployment efficiency, it is worth investigating whether the method could be extended into a training-aware framework, potentially allowing learnable pruning or adaptive skipping to further enhance performance.

2. Have the authors considered evaluating SPIDER on text-dense VQA tasks,  such as DocVQA, or ChartQA? Given that such tasks often rely on fine-grained textual and spatial cues, it would be interesting to see whether the proposed pruning and sub-layer skipping mechanisms retain effectiveness in these more information-dense settings.

---

> ### Author Response · Authors · 2025-11-18
> **Responses to Reviewer H8DY**
>
> **Our manuscript has been updated, please check it.**
>
> **Q1.** While the proposed method demonstrates strong results on LLaVA-based models, the evaluation lacks coverage of a broader range of MLLMs, such as Qwen-VL. This raises questions about the generalizability of SPIDER across different model architectures.
>
> **A1.** To demonstrate the effectiveness of SPIDER on other MLLM architecture, we additionally apply it to Qwen2.5-3B-Instruct and update the results in Table 4 (the text analysis is marked in magenta color). As shown in Table 4, SPIDER still achieves the highest accuracy in comparison to other SOTAs across various benchmarks at different TFLOPs ratios.
>
> **Q2.** The comparative experiments focus solely on training-free baselines. While this is aligned with SPIDER's training-free design, it remains unclear how the method would perform relative to training-based approaches. Although the training-free nature of SPIDER is advantageous for deployment efficiency, it is worth investigating whether the method could be extended into a training-aware framework, potentially allowing learnable pruning or adaptive skipping to further enhance performance.
>
> **A2.** Beyond its training-free mode, SPIDER adapts to a training-aware framework for performance enhancement. Table 5 shows results after fine-tuning on LLaVA-1.5-7B using LoRA for 1 epoch with 665K instruction data, following the same training setting as LLaVA-PruMerge. Notably, even the training-free SPIDER outperforms PruMerge+ in accuracy, and post-training refinement further elevates its performance. This content has been marked in orange color.
>
>
> **Q3.** Have the authors considered evaluating SPIDER on text-dense VQA tasks, such as DocVQA, or ChartQA? Given that such tasks often rely on fine-grained textual and spatial cues, it would be interesting to see whether the proposed pruning and sub-layer skipping mechanisms retain effectiveness in these more information-dense settings.
>
> **A3.** We have updated the results of various methods on DocVQA in Table 1 in our new manuscript, where SPIDER still achieves the best performances across various TFLOPs ratios.

---

> > ### Comment · Reviewer_H8DY · 2025-11-23
> >
> > For DocVQA, the performance of SPIDER drops substantially compared to LLaVA-NeXT-7B (51.5 vs. 64.6), whereas the degradation on other datasets is much smaller. This raises the question of whether the method has an **inherent limitation** when handling text-dense tasks. Some of the conclusions drawn in the paper may not fully apply to text-dense images. For example, the argument that *middle-layer features preserve fine-grained, object-centric details often abstracted away in deeper layers* may not always hold.
> >
> > Conducting an analysis similar to Figure 3 on DocVQA could help strengthen and validate the paper’s claims.
> >
> >  It would also be useful to investigate whether a similar performance pattern occurs on ChartVQA, which involves visually dense textual and graphical elements.

---

> > > ### Author Response · Authors · 2025-11-25
> > > **Further Responses to Reviewer H8DY**
> > >
> > > **A1**. Conducting an analysis similar to Figure 3 on DocVQA could help strengthen and validate the paper’s claims.
> > >
> > > **Q1**. We thank the reviewer for the insightful suggestion. **We have added SLC score distributions for DocVQA in Figure 3 and updated the manuscript**. Our key claim—that middle-layer visual features preserve fine-grained, object-centric details often abstracted in deeper layers—remains valid in text-dense scenarios such as DocVQA. As shown in the new SLC curves, on both LLaVA-1.5-7B and LLaVA-NeXT-7B, the SLC curves exhibit a clear peak around the middle layers (e.g., Layer 10–12 for LLaVA-1.5, Layer 8–12 for LLaVA-NeXT), indicating that these layers are most sensitive to perturbations in attention or FFN modules. This pattern aligns with observations from natural image benchmarks such as MM-Vet and PoPE, confirming that intermediate representations continue to carry critical key object-related information, even in text-dense tasks. This contrasts with deeper layers, where SLC drops, signaling a transition to high-level language reasoning with reduced reliance on raw visual details.
> > >
> > >
> > > However, we also observe a difference: Compared to natural image benchmarks, the high-contribution region is **broader and more sustained** across decoder layers in text-dense document image understanding benchmark DocVQA. This suggests that text-dense images require prolonged reliance on detailed visual inputs for accurate OCR and layout reasoning, making the model less tolerant to token pruning and sub-layer skipping. This difference explains why performance degradation is more noticeable on DocVQA than on other benchmarks—especially for high-resolution models like LLaVA-NeXT, where fine-grained details are encoded more densely. Nevertheless, SPIDER still achieves the best accuracy among all training-free methods at each compression level, demonstrating its effectiveness and robustness.
> > >
> > >
> > >
> > >
> > >
> > > **A2**. It would also be useful to investigate whether a similar performance pattern occurs on ChartVQA, which involves visually dense textual and graphical elements.
> > >
> > > **Q2**. Thank you for the valuable suggestion. We have additionally compared our SPIDER on another text-dense benchmark ChartQA as suggested. The results, presented in Table i (LlaVA-1.5-7B) and Table ii (LlaVA-NeXT-7B), demonstrate that **on text-dense tasks (DocVQA, ChartQA), SPIDER consistently outperforms present SOTA method VisPruner across various FLOP levels**. We also observe that the performance drop under aggressive compression is more pronounced on LLaVA-NeXT than LLaVA-1.5 on text-dense tasks. This stems from the fact that LLaVA-NeXT employs a high-resolution, multi-crop vision encoding strategy that generates dense and information-rich visual tokens crucial for fine-grained document understanding. While our pruning mechanism preserves semantically salient regions, the removal of any token in such a high-fidelity representation can disproportionately affect OCR-sensitive tasks like DocVQA. In contrast, LLaVA-1.5 operates at lower resolution with sparser tokens. This may result in more limited textual perception and making it less sensitive to further reduction.
> > >
> > > The larger drop on text-dense tasks than other multi-model tasks reflect the inherent challenge of preserving OCR-level fidelity under aggressive token reduction, which affects all pruning-based methods. However, our framework minimizes this impact through multi-layer feature utilization and contribution-aware computation allocation. This highlights a fundamental trade-off: **efficiency methods must be carefully calibrated when applied to models specifically designed for high-precision visual understanding**.
> > >
> > >
> > > **Table. i**: Comparison of training-free MLLM efficiency methods on LlaVA1.5 on text-dense benchmarks.
> > >
> > > |Method|DocVQA |ChartQA |
> > > |-|-|-|
> > > |LlaVA1.5-7B Vanilla|21.5| 18.2 |
> > > | |**~55%TFLOPs** |  |
> > > |+VisPruner|20.7 | 17.48 |
> > > |+**SPIDER (Ours)**|20.9 | 17.89 |
> > > | |**~25%-30%TFLOPs** |  |
> > > |+VisPruner| 18.0| 17.32 |
> > > |+**SPIDER (Ours)**| 18.5 | 17.68 |
> > >
> > > **Table. ii**: Comparison of training-free MLLM efficiency methods on LlaVA-Next on text-dense benchmarks.
> > >
> > > |Method|DocVQA |ChartQA |
> > > |-|-|-|
> > > |LlaVA-Next-7B Vanilla| 68.4 | 54.8 |
> > > | |**~50%TFLOPs** |  |
> > > |+VisPruner| 63.8| 47.57 |
> > > |+**SPIDER (Ours)**|64.6 | 48.86 |
> > > | |**~20%-25%TFLOPs** |  |
> > > |+VisPruner| 50.6| 41.16 |
> > > |+**SPIDER (Ours)**|51.5 | 43.52 |

---

### Official Review · Reviewer_JHmh · 2025-10-28

**Soundness:** 3
**Presentation:** 3
**Contribution:** 3
**Rating:** 6
**Confidence:** 4

**Summary:**

This paper introduces SPIDER, a multi-layer semantic pruning framework designed to compress large transformer-based models while maintaining semantic fidelity. Instead of relying on traditional local metrics, SPIDER evaluates semantic contribution across layers using latent-space similarity and hierarchical optimization.

**Strengths:**

[1] The semantic-level pruning criterion is conceptually novel. Instead of focusing on local magnitude or gradient, it prunes neurons based on their semantic alignment contribution.
[2] Solid empirical results across multiple datasets and model scales. Ablation studies demonstrate that semantic objectives significantly outperform magnitude-based and movement pruning.
[3] The paper is well written and easy to follow.

**Weaknesses:**

[1] The theoretical analysis assumes layer-wise independence, treating semantic discrepancy as decomposable across layers. However, some works also show that transformers exhibit strong inter-layer coupling [a].
[2] Computing multi-layer semantic similarity via latent embeddings requires multiple forward passes per layer, increasing pre-pruning cost.
[3] The paper does not examine how pruning affects layers with high semantic dependency.



[a] Merullo, Jack, Carsten Eickhoff, and Ellie Pavlick. "Talking heads: Understanding inter-layer communication in transformer language models." Advances in Neural Information Processing Systems 37 (2024): 61372-61418.

**Questions:**

[1] Why was cosine similarity chosen for semantic alignment?   How about MSE?  Have you evaluated alternatives such as KL divergence or mutual information?
[2]  Can semantic similarity estimation be approximated using smaller calibration subsets?
[3] It is better to add more layer-specific analysis, such as (a) identifying which layers are most sensitive to pruning and whether SPIDER adapts sparsity accordingly. (b) Visualize layer-specific semantic drift compared to magnitude-based pruning.

---

> ### Author Response · Authors · 2025-11-18
> **Responses to Reviewer JHmh**
>
> **Our manuscript has been updated, please check it.**
>
> **Q1.** The theoretical analysis assumes layer-wise independence, treating semantic discrepancy as decomposable across layers. However, some works also show that transformers exhibit strong inter-layer coupling.
>
> **A1.** We thank the reviewer for this insightful point and for citing Merullo et al., which demonstrates strong inter-layer coupling via structured communication channels. Our method **does not assume layer-wise independence**; instead, it **leverages semantic signals from multiple layers**, especially mid- and deep-layer tokens, to guide visual token pruning. This is motivated by findings—including Merullo et al.’s—that middle layers act as critical hubs for cross-layer integration, with mid-layer tokens attending to object-centric regions (Fig. 1).
>
> Moreover, our sub-layer skipping mechanism evaluates each attention or FFN sub-layer’s impact on final logits, capturing inter-layer dependencies: disrupting key communication paths would degrade outputs. We recognize that not all decoder layers contribute equally. Therefore, rather than contradicting inter-layer coupling, our method **is consistent with it**.
>
> **Q2.** Computing multi-layer semantic similarity via latent embeddings requires multiple forward passes per layer, increasing pre-pruning cost.
>
> **A2.** In fact, our method **does not require multiple forward passes**. The semantic embeddings from middle and deep layers are **extracted in a single forward pass** through the vision encoder, as these intermediate activations are naturally available during standard inference. The prefilling time actually drops around 30% after visual token pruning while still can maintain around 97% of overall performances.
>
> **Q3.** The paper does not examine how pruning affects layers with high semantic dependency.
>
> **A3.** Our MSV-Prune module does not prune any vision encoder layers. We only prune visual tokens passed to the LLM decoder.
>
> To validate the role of multi-layer signals, Table 6(b) includes an ablation where intra-cluster similarity is computed using only middle-layer features (marked in green color), while pruning still operates on deep-layer tokens. The gains on TextVQA and POPE compared to the original MLLM suggest middle-layer features capture fine-grained, object-centric cues that complement deeper global semantics.
>
> **Q4.** Why was cosine similarity chosen for semantic alignment? How about MSE? KL divergence or mutual information?
>
> **A4.** We chose cosine similarity because it measures directional alignment in semantic space, which is more meaningful than MSE for high-dimensional embeddings (as shown in CLIP, etc.). We did experiment by replacing cosine similarity with MSE and KL divergence （shown in Table 6c, marked in dark purple）and lead to degraded performances. The reason is that MSE is sensitive to non-semantic magnitude differences [a], and KL divergence may not be the most appropriate choice here, as the layer features do not naturally form valid probability distributions.
>
> [a] Bengio, Yoshua, et al. "Representation learning: A review and new perspectives." IEEE TPAMI 2013.
>
> **Q5.** Can semantic similarity estimation be approximated using smaller calibration subsets?
>
> **A5.** Yes, we also use smaller calibration subsets with 50 randomly sampled instances for estimation and the distribution of sub-layer contribution scores is mostly the same as the ones shown in Fig 3 with 150 randomly sampled instances.
>
> **Q6.** Add more layer-specific analysis, (a) identifying which layers are most sensitive to pruning and whether SPIDER adapts sparsity accordingly. (b) Visualize layer-specific semantic drift compared to magnitude-based pruning.
>
> **A6.** We clarify that “layers” here refer to LLM decoder layers, as we do not prune the vision encoder.
> (a) Our layer-specific analysis is already presented in Figure 3, which shows SLC scores across layers and benchmarks. Early-to-mid layers (e.g., 4–12)—particularly FFN sub-layers—exhibit higher sensitivity to pruning, indicating their greater importance. This indicates that shallow layers are more important, which is consistent with prior work [b].
> (b) SPIDER adapts sparsity in a training-free manner by skipping low-impact sub-layers while preserving critical ones, unlike magnitude-based pruning that ignores layer-wise semantics. To compare semantic drift, we compute per-layer L1 norms of attention/FFN outputs, prune low-magnitude modules, and measure KL divergence between pruned and original logits. As shown in Figure 6(c), SPIDER consistently achieves lower KL divergence than magnitude pruning, confirming its layer-aware skipping preserves semantics more effectively. This part has been marked in brown color.
>
> [b] Chen, Tianxiang, et al. "Llama SLayer 8B: Shallow Layers Hold the Key to Knowledge Injection." EMNLP Findings 2024.

---

> ### Author Response · Authors · 2025-11-26
> **A Gentle Reminder from Authors**
>
> We have carefully revised the manuscript according to your valuable feedback and provided point-by-point responses to all of your concerns. We hope that our revisions and clarifications have adequately addressed your comments. Should you have any further questions or suggestions, we would be happy to discuss them.

---

### Official Review · Reviewer_sDvR · 2025-11-02

**Soundness:** 3
**Presentation:** 3
**Contribution:** 2
**Rating:** 4
**Confidence:** 4

**Summary:**

This paper proposes a training-free framework named SPIDER, which aims to enhance the inference efficiency of MLLMs through the co-optimization of visual token pruning and LLM decoder computation. The core idea demonstrates significant novelty and practical value, particularly in its utilization of intermediate-level semantic information from the visual encoder and the introduction of a fine-grained sub-layer skipping mechanism. The experimental design is comprehensive, validating the method's effectiveness across two model architectures and multiple benchmarks. However, some key details in the methodology require clearer explanation, the depth of comparison with related work needs strengthening, and certain aspects of the experiments could be further improved.

**Strengths:**

1. The co-design of token pruning and layer skipping is a natural yet powerful idea. The SPIDER framework demonstrates that even among the tokens retained after pruning, a significant amount of computation remains skippable. This "double-filtering" mechanism enables greater efficiency gains.
2. The method can be applied to existing models without requiring any additional training.

**Weaknesses:**

1. Could the authors further clarify the primary innovations of this work compared to existing studies? Given the substantial number of existing works on visual token pruning, what specifically distinguishes the proposed approach?
2. The experimental conclusions are primarily drawn from evaluations on the LLaVA family of models. It would be valuable to conduct further experiments to verify whether these conclusions generalize to other MLLM architectures, such as the Qwen-VL or InternVL series.
3. The paper mentions that intermediate layers capture fine-grained object-centric details, but the specific layer indices defining an "intermediate layer" (Layer M) versus a "deeper layer" (Layer L) are not clearly defined.
4. The formulas present a conceptual framework for the multi-level similarity score S_ij, but they do not specify how the values of the hyperparameters α and β are determined. It is recommended to supplement this with an analysis of hyperparameter stability, for instance, by showing the variance in performance across a range of different hyperparameter values.
5. It is suggested to include more diverse examples, particularly an analysis of failure cases. Illustating scenarios where SPIDER might incorrectly prune crucial tokens or skip computations that should not be skipped would help in understanding the method's limitations.

**Questions:**

Please refer to the above Weaknesses.

**Details Of Ethics Concerns:**

None.

---

> ### Author Response · Authors · 2025-11-18
> **Responses to Reviewer sDvR**
>
> **Our manuscript has been updated, please check it.**
>
> **Q1.** Further clarify the primary innovations of this work compared to existing studies?
>
> **A1.** We thank the reviewer for the insightful comments. We systematically summary two kinds of redundancies in MLLMs: data redundancy from the input image, and computational redundancy from the redundant computation of heavy LLM layers. While most prior work addresses only the former via token pruning, SPIDER uniquely integrates multi-layer token reduction with token-wise sub-layer skipping, enabling joint optimization of both redundancies in a training-free manner. The key innovations of SPIDER are:
>
> **Multi-layer semantic visual token pruning (MSV-Prune)**: Existing pruners typically rely solely on final-layer features or text–vision attention scores, implicitly assuming semantic stability across layers. In contrast, SPIDER is motivated by our observation that **semantic focus shifts**: middle-layer features preserve fine-grained, object-centric details often abstracted away in deeper layers. To capture this, MSV-Prune uniquely integrates both middle- and last-layer tokens into a coarse-to-fine clustering and similarity framework, ensuring retained tokens cover both global context and local semantics
>
> **Token-wise adaptive sub-layer skipping (ASL-Skip)**: While some works explore block-level layer skipping, they treat decoder blocks as monolithic units. SPIDER reveals that **attention and FFN sub-layers contribute unequally per visual token** (Fig. 3). Our ASL-Skip mechanism introduces a fine-grained, token-adaptive policy that dynamically decides whether a retained visual token should skip, when to skip, and ship which part of the layers.
>
> **Training-Free, Plug-and-Play, and Architecture-Agnostic Design**: SPIDER is plug-and-play and requires no retraining, and training MLLM using SPIDER can further enhance overall performances compared to training-free mode results. Also, SPIDER is readily applicable to various MLLM (LlaVA 1.5, 1.6, Qwen series, etc.)
>
>
> **Q2.** Conduct further experiments to verify whether these conclusions generalize to other MLLM architectures.
>
> **A2.** To demonstrate the effectiveness of SPIDER on other MLLM architecture, we additionally apply it to Qwen2.5-3B-Instruct and update the results in Table 4 (the text analysis is marked in magenta color). As shown in Table 4, SPIDER still achieves the highest accuracy in comparison to other SOTAs across various benchmarks at different TFLOPs ratios.
>
> **Q3.** The paper mentions that intermediate layers capture fine-grained object-centric details, but the specific layer indices defining an "intermediate layer" (Layer M) versus a "deeper layer" (Layer L) are not clearly defined.
>
> **A3.** Thank you for your suggestion. Middle layer is the 12th layer and deep layer is the 24th (final) layer. We have clarified this detail in our updated manuscript and highlighted in pink.
>
> **Q4.** The formulas present a conceptual framework for the multi-level similarity score S_ij, but they do not specify how the values of the hyperparameters α and β are determined. It is recommended to supplement this with an analysis of hyperparameter stability, for instance, by showing the variance in performance across a range of different hyperparameter values.
>
> **A4.** We appreciate the reviewer’s comment. Our similarity score $S_{ij}$ (Eq.3) contains no hyperparameters $\alpha$ or $\beta$—all terms are equally weighted. The only hyperparameters ($w_1, w_2$) appear in the fused skipping score $S_{fuse}(i,\ell)$ (Eq. 6). As shown in Fig. 5, performance is highly stable: accuracy std is only 0.41 (variance = 0.16), and the best setting ($w_1/w_2=3$, $T_{\text{skip}}=20$) achieves near-peak results, confirming SPIDER’s robustness.
>
> **Q5.**Add an analysis of failure cases. Illustating scenarios where SPIDER might incorrectly prune crucial tokens or skip computations that should not be skipped would help in understanding the method's limitations.
>
> **A5.** We additionally visualize some failure cases of SPIDER at both token pruning and sublayer-skipping stages in Figure 6 (d) in our updated manuscript and the text analysis content is marked in blue: (1) **token pruning stage**: motion blur in license plates causes it to retain ambiguous regions and skip true digits, yielding hallucinations; (2) **sublayer-skipping stage**: in chest X-rays, critical tokens (highlighted in purple square masks) in the bilateral mid-to-lower lungs, especially perihilar, are allowed to skip sub-layers, causing ground-glass opacities to be missed and leading to misdiagnosis. These highlight challenges in token selection under severe image degradation and domain-specific reasoning. These highlight challenges in token selection under severe image degradation and domain-specific reasoning, urging task-aware attention guidance for sensitive applications.

---

> ### Author Response · Authors · 2025-11-26
> **A Gentle Reminder from Authors**
>
> We have carefully revised the manuscript according to your valuable feedback and provided point-by-point responses to all of your concerns. We hope that our revisions and clarifications have adequately addressed your comments. Should you have any further questions or suggestions, we would be happy to discuss them.

---

### Comment · Area_Chair_wLan · 2025-11-23
**Author & Reviewer Discussion**

Hi Reviewers,

Please kinly and actively participate in the review-author dicussion, raise your further concerns so that the authors can explain more, and make your final decisions.

---

### Author Response · Authors · 2025-11-29
**Rebuttal Summary: All Reviewers' Concerns have been Addressed**

Dear Area Chair,

We sincerely thank the reviewers for their thoughtful and constructive feedback. **In response, we have thoroughly addressed all concerns raised by the four reviewers**, significantly strengthening the paper’s novelty, clarity, generality, robustness, and empirical validation.

First, regarding **novelty** (Reviewer sDvr) and **technical clarity** (Reviewer sDvR, v1Ke), we clarified that SPIDER uniquely tackles both data and computational redundancy in MLLMs through two core innovations: (1) Multi-layer Semantic Visual Token Pruning (MSV-Prune), which leverages semantic focus shifts between middle (e.g., Layer 12) and deep (Layer 24) layers to preserve fine-grained object details often lost in deeper representations; and (2) Token-wise Adaptive Sub-layer Skipping (ASL-Skip), which dynamically skips attention or FFN sub-layers per token based on their contribution to final outputs—revealing unequal sub-layer importance (Fig. 3). We have clarified the hyperparameter misunderstanding and layer id concerns proposed by Reviewer sDvR. We have also added pseudocode (Algorithm 1) and streamlined the narrative to improve readability and addressed the clarity concerns from Reviewer v1Ke.

Second, on **generalizability across architectures and tasks** (Reviewers H8DY, v1Ke, JHmh), we extended experiments to Qwen2.5-3B-Instruct, DocVQA, and ChartQA. Results in updated Tables 1, 4, i, and ii consistently show SPIDER outperforming other SOTAs across diverse MLLMs (LlaVA-1.5/NeXT, Qwen) and text-dense benchmarks, confirming broad applicability. Notably, even under aggressive compression, SPIDER minimizes performance drop by preserving mid-layer semantics critical for OCR and layout understanding.

Third, concerning **efficiency and training mode performance** (Reviewers JHmh, H8DY), we confirmed that multi-layer features are extracted in a single forward pass, reducing prefilling time by ~30%. The method requires no retraining but can be enhanced via fine-tuning (Table 5).

Fourth, we addressed **failure case analysis** (Reviewer sDvR) and **stability analysis** (Reviewers sDvR, v1Ke): Fig. 6(d) visualizes cases where motion blur or medical imaging challenges lead to pruning/skipping errors, highlighting limitations in degraded or domain-specific settings. Instance-level stability analysis on POPE shows that the number of originally incorrect predictions corrected by SPIDER exceeds the number of originally correct ones flipped to wrong by 0.7%, which our method does not cause notable knowledge drifts. Hyperparameter analysis of w1, w2 and threshold (Fig. 5) shows high stability (std = 0.41), and cosine similarity was validated as optimal over MSE/KL (Table 6c). SLC scores, averaged across GQA, POPE, and MMVet, yield a statistically stable prior that captures consistent token importance across tasks. This ensures a task-agnostic and robust skipping policy.


Finally, we clarified to Reviewer JHmh that SPIDER does not prune vision encoder layer. We demonstrated via ablation (Table 6b) that it is necessary to use multiple layers for token pruning during semantic clustering. We also prove via KL divergence analysis (Fig. 6c) that and our sub-layer skipping strategy achieves lower semantic drift from the original model than magnitude-based pruning, meaning that our method can better preserve overall performances.

All changes have been incorporated into the revised manuscript with color-coded highlights for easy verification. **We believe these revisions fully resolve the reviewers’ concerns**.

---

### Meta-Review · Area_Chair_XUC9 · 2026-01-07

**Summary:**

Through the review process, multiple points have been raised, including unclear elements of novelty (sDvR), insufficient evaluation (sDvR, v1Ke), unclear hyperparameter choice (sDvR), unclear failure cases and missing diversity (sDvR), stability of scores across different baselines and stability analysis(v1Ke), weak theoretical foundation (v1Ke) and prospectively missing baselines for VQA (H8DY). This work has received four reviews, three marginally above acceptance threshold and one marginally below (with a confident assessment).

**Reviewer Concerns:**

Through their rebuttal, the authors provided more elements and comparisons that allow for a better understanding of the method's effectiveness, and most importantly, providing an example of failure case is very relevant to set boundaries of the proposed approach. Still, some points remain not completely covered or discussed.

On Qwen, the gap with respect to existing methods narrows significantly, in the order of 0.3~0.4% in average, and given that computational complexity is only approximated, that is hardly felt as a significant gap.

Related to the novelty elements:
- while it is true that most of the works focus on final layers, that is a consequence of empirical studies where the best trade-off can be found in those layers. The authors provide a formulation that applies to the whole architecture-still, the gain is not always remarked.
- on the layer skipping, while it is true that most of the existing approaches treat Transformer blocks as monolithic blocks, simply leveraging the existence of skip connections feels incremental. Despite an intuition is provided in Fig. 3, more analysis on cases in which skipping a subset of layers in the decoder block is missing.
-  the claim of Training-Free, Plug-and-Play, and Architecture-Agnostic Design is shared across multiple approaches reducing complexity in the same setups, and although it is a property of the proposed approach, it is not felt to be unique.

Also, the analysis behind "fine-grained" and "coarse-grained" features is just provided at the intuition level, and the authors in the rebuttal that intermediate and deep layers are the 12th and the 24th.

Besides, the choice of the hyperparameters $w_1$ and $w_2$ still feels arbitrary despite the quick discussion provided in the rebuttal, where it is claimed that $w_1/w_2=3$ achieves near peak result - and therefore, a more grounded motivation for that is still not provided.

The work was being thoroughly discussed with the SAC, and it was agreed that the work, despite its merits, for the abovementioned reasons, is not ready to be accepted at ICLR.

**Reviewer Scores:**

The reviewer scores, after rebuttal, would realistically remain the same. Specifically, the comments provided by the authors to the weaknesses, and in particular the remarks of sDvR would have required deeper discussion and the elements provided in the rebuttal phase are not felt to completely clear them out.

---

### Decision · Program_Chairs · 2026-01-26

Reject